# Study of the Printability, Microstructures, and Mechanical Performances of Laser Powder Bed Fusion Built Haynes 230

**Ziheng Wu** *,†, **Srujana Rao Yarasi**, **Junwon Seo**, **Nicholas Lamprinakos** and **Anthony D. Rollett** *

Department of Materials Science & Engineering, Carnegie Mellon University, 5000 Forbes Ave,
Pittsburgh, PA 15213, USA
* Correspondence: wu57@llnl.gov (Z.W.); rollett@andrew.cmu.edu (A.D.R.);
  Tel.: +1-765-418-5254 (Z.W.); +1-412-268-3177 (A.D.R.)
† Current affiliation: Materials Engineering Division, Lawrence Livermore National Laboratory, 7000 East Ave,
  Livermore, CA 94550, USA.

**Abstract:** The nickel-based superalloy, Haynes 230 (H230), is widely used in high-temperature applications, e.g., heat exchangers, because of its excellent high-temperature mechanical properties and corrosion resistance. As of today, H230 is not yet in common use for 3D printing, i.e., metal additive manufacturing (AM), primarily because of its hot cracking tendency under fast solidification. The ability to additively fabricate components in H230 attracts many applications that require the additional advantages leveraged by adopting AM, e.g., higher design complexity and faster prototyping. In this study, we fabricated nearly fully dense H230 in a laser powder bed fusion (L-PBF) process through parameter optimization. The efforts revealed the optimal process space which could guide future fabrication of H230 in various metal powder bed fusion processes. The metallurgical analysis identified the cracking problem, which was resolved by increasing the pre-heat temperature from 80 °C to 200 °C. A finite element simulation suggested that the pre-heat temperature has limited impacts on the maximum stress experienced by each location during solidification. Additionally, the crack morphology and the microstructural features imply that solidification and liquation cracking are the more probable mechanisms. Both the room temperature tensile test and the creep tests under two conditions, (a) 760 °C and 100 MPa and (b) 816 °C and 121 MPa, confirmed that the AM H230 has properties comparable to its wrought counterpart. The fractography showed that the heat treatment (anneal at 1200 °C for 2 h, followed by water quench) balances the strength and the ductility, while the printing defects did not appreciably accelerate part failure.

**Keywords:** laser powder bed fusion; Ni superalloy; hot cracking; high-temperature creep; tensile property; fractography





## 1. Introduction

The use of laser powder bed fusion (L-PBF) has grown drastically in the past decades. As one of the metal additive manufacturing (AM) technologies with high technical maturity, L-PBF-built components are widely used. L-PBF uses layer-by-layer build sequence, fine feedstock size (~10–60 μm), and accurate laser, which enable the fabrication of complex geometries and reduce material wastes. More importantly, our application exploited its ability for fast prototyping and reducing the turnaround time for design optimization. The target application is the primary heat exchanger in Generation 3 concentrated solar power [1], which operates at high pressure (200 bar) and high temperature (720 °C) in a molten salt corrosive environment and with daily cyclic loading. Haynes 230 (H230) is one of the candidate alloys that offers exceptional high-temperature strength and stability, as well as corrosion resistance. Yet, as with many other alloys that were originally designed for conventional metal processing techniques, there are few data on its printability or the mechanical performance of the corresponding AM-built components.

H230 is an austenitic solid-solution strengthened nickel-based superalloy. The alloying elements Cr and W introduce a local stress field in the FCC $\gamma$ matrix that impedes dislocation motion, thus increasing the alloy strength. The alloy was designed for high-temperature applications; so, it was expected to maintain its mechanical performance at our service temperature of 720 °C. Haynes International published a detailed report [2] on the mechanical performance of H230 over a wide range of temperatures (often up to 1000 °C) for long testing periods (often up to 1000 h). The report also demonstrated the exceptional thermal stability of the H230 microstructure by showing the room temperature tensile testing results of samples with long thermal exposure, e.g., 760 °C for 50,000 h. Veverkova et al. [3,4] evaluated the thermal stability of H230 from 600 °C to 1050 °C; they found that H230 maintained its tensile properties and microstructure after a long exposure at 850 °C.

Hrutkay and Kaoumi [5] tested the tensile strengths of H230 over a wide range of temperatures (25–950 °C), and they showed that strain rate sensitivity ($10^{-3}$–$10^{-5}$·s$^{-1}$) gradually emerged only above 800 °C. This indicated the combined effects of the hardening and softening and implied an upper limit for the operating temperature. Boehlert et al. [6] assessed the creep performance and the microstructure of cold rolled H230 at 700 °C, 760 °C, and 815 °C. Despite the differences in microstructure, i.e., coarser grains and carbides, the time-dependent creep curves in Boehlert's study provided important references by suggesting that H230 maintains slow and steady state creep rates for the testing period. Pataky et al. [7] identified dislocation climb and solute drag as the creep rate limiting mechanisms for H230 at 800 °C and 900 °C. With the assistance of digital image correlation, they suggested that the triple junctions facilitated void formation and contributed to creep damage. Yoon et al. [8] used aging treatments to introduce serrated grain boundaries (GBs) and limit the formation of lamellar carbides. They suggested that microstructural stability at 900 °C was improved as the serrated grain boundaries were more resistant to cavitation creep damage. Jiang et al. [9] found that higher solution temperatures and longer solution times increased the amplitude of the grain boundary serrations.

In addition to pure creep, the creep–fatigue interaction also affects the service life of the material when cyclic loading is present. Lu et al. [10] studied the effects of the test temperature and the hold time on crack growth behaviors. Both higher temperatures and longer hold times led to the transgranular–intergranular transition on the fracture appearance. Choi [11] assessed the creep–fatigue–ratcheting interactions of H230 with multi-axial loading for temperatures from 649 °C to 982 °C. The study indicated that more factors influenced the material failure mechanisms as the temperature increased. The different cyclic responses of H230 were significant with only 100 °C separation.

Unlike the widely printed Inconel 718, the available literature for the AM-built H230 is limited. Back in 2015, Bauer et al. [12] showcased a successful H230 build in L-PBF which resulted in nearly fully dense part densities and tensile properties comparable to those of the wrought materials. Haack et al. [13] built H230 via laser metal deposition; the study evaluated several heat-treating procedures and showed that proper aging treatments led to grain boundary serrations which could be beneficial to the mechanical properties. Yang et al. [14] conducted a comprehensive study of the strengthening mechanisms of L-PBF H230. The study revealed numerous carbide-matrix orientation relationships and showed the carbide evolution from $M_{23}C_6$ to well-aligned $M_6C$ after stress relief. Xia et al. [15] evaluated the microstructure and mechanical properties of carbide-reinforced H230 prepared by selective laser melting and showed that the high carbide contents resulted in a tradeoff between tensile strength and ductility.

Cracking is a persistent problem for processing superalloys, especially in AM which normally has higher cooling rates and local stress gradients. It is one of the important printability issues that limit the availability of AM high-temperature alloys. Cracking has been reported on many nickel-based alloys such as Hastelloy-X [16,17], Haynes 282 [18], Inconel 738 [19], CM247LC [20], etc. Since the 1950s, extensive research has attempted to explain the mechanisms and develop models to predict crack formation [21–23].

Depending on the solidification stage when the nucleation and propagation occur, the cracking mechanisms can be categorized as (1) solidification cracking, (2) liquation cracking, (3) strain-age cracking (SAC), and (4) ductility-dip cracking (DDC). The first two mechanisms are associated with the liquidus. Solidification cracking occurs in the mushy zone when the volume fraction of the solid is above 85–95%. Cracks originate in the interdendritic regions where the dendrite formation inhibits the liquid backfilling. Liquation cracking is found in the heat affected zone where certain microsegregation-induced boundary phases with lower liquidus temperatures, e.g., low melting point carbides, are selectively remelted. SAC and DDC often correspond to reheating and aging treatments. When the precipitation-induced stress outweighs the stress relaxation, SAC may occur. Young et al. [24] suggested that DDC is a similar mechanism to SAC, which relates to precipitation and ductility reduction in the intermediate temperature conditions. Lippold and his colleagues [25–28] argued that DDC is a "creep-like" mechanism that is caused by the void formation due to grain boundary sliding and the lack of dynamic recrystallization. Tang et al. [29] performed an analysis of multiple alloy systems and concluded that both end-of-solidification behavior and high temperature toughness affect the tendency to hot cracking. Cracking is essentially a competition between the formation and the healing process during solidification. Weak links and residual stress are the key elements for crack propagation. The interdendritic/intercellular microfissures and liquid films and the boundary voids are high stress concentrators, which are subjected to thermal shrinkage and propagation under the influence of residual stress at cooling. Factors such as freezing range, precipitation, high temperature strength, and cooling rate can influence the cracking behaviors.

Despite the absence of $\gamma'$ and $\gamma''$, H230 is still susceptible to cracking, especially in AM, which adds additional difficulties to solving this problem due to the complex thermal history, the unique microstructures, and the stress distribution caused by the varying scanning strategy. Surprisingly, the aforementioned studies [12–14] registered no cracks in the as-built H230 specimens. The marginal difference in the alloying composition and the printing conditions could have affected crack severity. In this study, the objective is to quantify and investigate the pores and the cracks found in the L-PBF-built H230. Through parameter optimization, we aimed to mitigate the defects and demonstrate that the AM H230 is comparable to its wrought counterpart in terms of part quality and mechanical properties.

## 2. Methods

### 2.1. Materials and Processing

The feedstock used in this study was gas atomized H230 powder provided by Haynes International (Kokomo, IN, USA). The powder size is −53/+15 μm, and the powder morphology can be found in Supplementary Figure S1. Table 1 shows the chemical compositions of the feedstock.

**Table 1.** Alloying compositions of the gas atomized H230 powder (wt.%).

| Ni | Cr | W | Mo | Mn | Si | Al | C | O | N | S |
|------|-------|-------|------|------|-------|------|-------|--------|--------|---------|
| 59.94 | 21.90 | 14.22 | 2.73 | 0.41 | 0.356 | 0.40 | 0.087 | 0.0117 | 0.0011 | <0.001 |

This study used an EOS M290 L-PBF machine (EOS GmbH, Krailling, Germany) to fabricate the testing coupons at a layer thickness of 40 μm and with a laser spot size of 100 μm. For parameter optimization, rectangular prisms (15 mm (L) × 10 mm (W) × 10 mm (H)) were fabricated in two separate builds at pre-heats of 80 °C and 200 °C, as shown in Figure 1a. Table 2 summarizes the different combinations of laser power, speed, and hatch spacing used for each prism. The tensile and the creep test specimens (Figure 1b) were built using the optimized parameters of H2, as shown in Table 2. The tensile testing and the specimen design follow the ASTM E8/E8M standard [30]. All the specimens, including the mechanical coupons, used the stripe scanning strategy, with a stripe width of 10 mm, a stripe overlap of 0.12 mm, and 67° pattern rotation between layers.

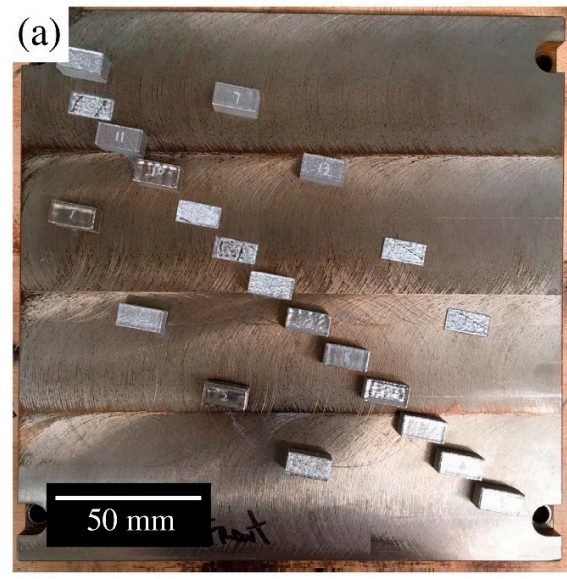

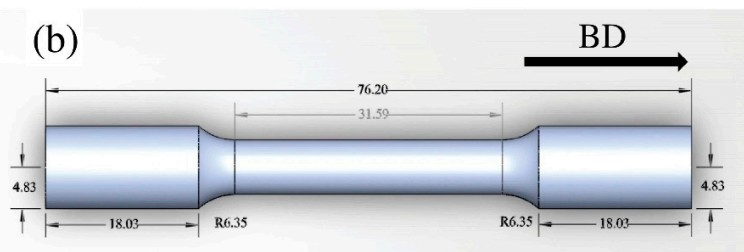

**Figure 1.** Photograph of L-PBF-built H230: (**a**) cube specimens for parameter optimization and (**b**) a schematic illustration of the tensile/creep test coupon with dimensions (unit: mm).

**Table 2.** Process parameters of the testing coupons. Note that all specimens share the same contour parameters. The prefix H stands for the high pre-heat temperature of 200 °C, and the prefix L stands for the low pre-heat temperature of 80 °C. The parameters for the mechanical coupons are highlighted in **bold**.

| | | | | Raster Infill | | | | | |
|---|---|---|---|---|---|---|---|---|---|
| Sample # | Pre-Heat (°C) | Power (W) | Velocity (mm/s) | Hatch (mm) | Sample # | Pre-Heat (°C) | Power (W) | Velocity (mm/s) | Hatch (mm) |
| L1/H1 | 80/200 | 285 | 700 | 0.11 | H15 | 200 | 330 | 1300 | 0.05 |
| L2/H2 | 80/**200** | **330** | **700** | **0.11** | H16 | 200 | 370 | 1300 | 0.05 |
| L3/H3 | 80/200 | 370 | 700 | 0.11 | H17 | 200 | 285 | 1600 | 0.05 |
| L4/H4 | 80/200 | 285 | 960 | 0.11 | H18 | 200 | 330 | 1600 | 0.05 |
| L5/H5 | 80/200 | 330 | 960 | 0.11 | H19 | 200 | 370 | 1600 | 0.05 |
| L6/H6 | 80/200 | 370 | 960 | 0.11 | H20 | 200 | 330 | 1900 | 0.05 |
| H7 | 200 | 285 | 400 | 0.11 | H21 | 200 | 370 | 1900 | 0.05 |
| H8 | 200 | 330 | 400 | 0.11 | L7 | 80 | 285 | 550 | 0.11 |
| H9 | 200 | 170 | 700 | 0.05 | L8 | 80 | 330 | 550 | 0.11 |
| H10 | 200 | 220 | 700 | 0.05 | L9 | 80 | 370 | 550 | 0.11 |
| H11 | 200 | 170 | 960 | 0.05 | L10 | 80 | 70 | 250 | 0.11 |
| H12 | 200 | 220 | 960 | 0.05 | L11 | 80 | 100 | 350 | 0.11 |
| H13 | 200 | 220 | 1300 | 0.05 | L12 | 80 | 130 | 450 | 0.11 |
| H14 | 200 | 285 | 1300 | 0.05 | | | | | |

| | Inner Contour | | | Outer Contour | |
|---|---|---|---|---|---|
| Power (W) | Velocity (mm/s) | Offset (mm) | Power (W) | Velocity (mm/s) | Offset (mm) |
| 139 | 390 | 0.012 | 80 | 800 | 0 |

All the as-built specimens were sectioned off from the build plate using a GF Machining Solutions AC Progress VP3 electrical discharge machine (EDM) (GF Machining Solutions, Biel/Bienne, Switzerland) before any subsequent treatments. The mechani-

cal coupons were annealed at 1200 °C for 2 h, followed by water quench. The primary concerns were the abrupt grain growth found at higher solution annealing temperatures, i.e., 1230–1280 °C [9] and the depletion of the solution strengthening elements due to the carbide formation/coarsening at just below the annealing temperature [2]. This typical heat treatment for H230 can prevent these detrimental effects, and meanwhile, stress relieve the as-built specimens.

## 2.2. Tensile and Creep Tests

The tensile and creep tests utilized the heat-treated specimens with the as-built surface finish. The as-built surface finish can better approximate the surface roughness of the heat exchangers, of which the prototyping efforts were supported by the mechanical testing in the current study. The room temperature tensile test was performed at a strain rate of $8.333 \times 10^{-5} \cdot s^{-1}$. Four repeat constant-load creep tests were performed at each of the two conditions of (a) 760 °C and 100 MPa and (b) 816 °C and 121 MPa in an ambient environment. The creep strain was continuously monitored, and the tests were set to stop at 500 h or sample rupture, whichever occurred first.

The selections of these test conditions took the rupture life and the availability of the creep data of the wrought H230 into consideration. The test condition (b) was selected as it was expected to cause severe creep damage within the testing time frame. As a result, the creep/failure mechanism could be identified in the fractography analysis.

## 2.3. Characterization and Testing

Quantification of the print defects was performed on the vertical EDM-sectioned surfaces, which were polished to a mirror finish following the steps of sandpapers down to 800 grits, 9 μm and 3 μm diamond paste, and 0.5 μm colloidal silica. To reveal the microstructure, the polished cross-section was electrolytically etched with 95% oxalic acid and 5% HCl before imaging using a Leica DM750M optical microscope (Leica Microsystems GmbH, Wetzlar, Germany).

An FEI Quanta 600 FEG (FEI Company, Hillsboro, OR, USA) and a Mira TESCAN MIRA3 FEG (TESCAN, Brno, Czechia) scanning electron microscope (SEM) were used to investigate how the heat treatments affected the microstructure and the fracture behavior. Backscattered electron (BSE) imaging and electron backscatter diffraction (EBSD) were used to reveal the microstructural texture and carbide precipitates, while secondary electron (SE) imaging was used to show the fracture surfaces of the tensile and the creep specimens. The specific imaging parameters vary, and they can be found in the corresponding image or figure caption.

## 2.4. Finite Element Stress Simulations

The finite element stress simulations with a voxel size of 0.1 mm$^3$ were run using ANSYS Additive 2020 R2 (Ansys Inc., Canonsburg, PA, USA). To better approximate the experiments, the simulations were performed on the geometry replicate of 15 (L) mm × 10 (W) mm × 10 (H) mm. The model adopted the room temperature material properties of H230 (see Table S1), the laser conditions of 330 W, 700 mm/s, and the hatch spacing of 0.11 mm, as well as the nominal scan strategy with 67° rotation as inputs to simulate the stresses developed during the cooling process. The analysis compared the maximum stress experienced by each location caused by the different pre-heat temperatures of 80 °C and 200 °C.

## 3. Results and Discussions

### 3.1. Printability of H230

Figure 2 maps the defect contents of the L-PBF H230 built at the 80 °C and 200 °C pre-heat temperatures. With a hatch spacing of 110 μm, laser powers above 285 W, and laser speeds above 600 mm/s, nearly fully dense specimens were produced. The energy densities of these corresponding parameter sets are between 67.5 J/mm$^3$ and 120.1 J/mm$^3$.

As the laser speed decreased to 550 mm/s or less, the defect contents increased owing to keyholing, which resulted in predominantly spherical keyhole pores. The results at the two pre-heat temperatures showed a matching trend of porosity variation; however, the cracking issue is the primary distinction between the two builds. Even within the optimal process space, the 80 °C pre-heat always resulted in a higher defect concentration at an identical parameter set compared with the 200 °C pre-heat. This study tested additional parameter sets with a smaller hatch spacing of 50 μm and similar energy densities at 200 °C for the purpose of mitigating the cracking issue. Figure 2 shows that these parameters also resulted in low defect concentrations.

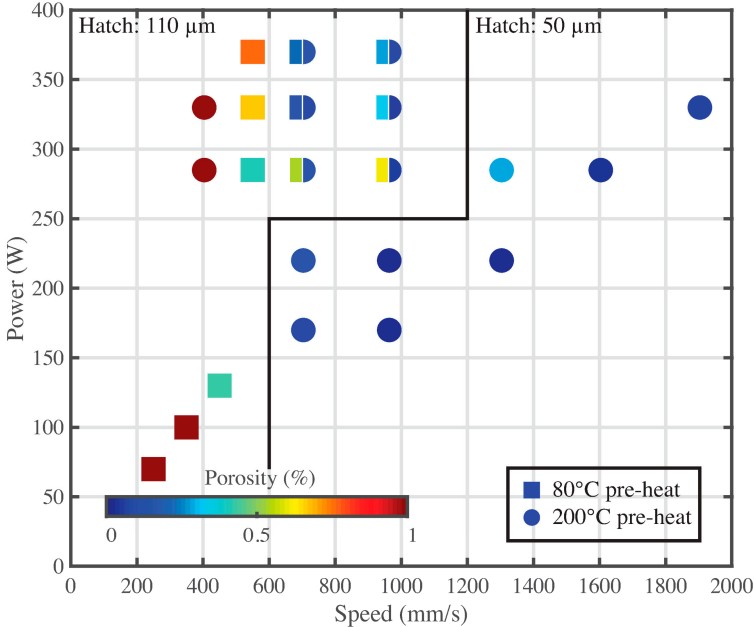

**Figure 2.** Process map showing the porosity contents in the rectangular prisms built using various laser powers, speeds, and hatch spacings at two different pre-heat temperatures of 80 °C and 200 °C. The optimal process space for H230 can be identified from the map. Note that a maximum porosity cutoff was set at 1% to highlight the porosity variations <1%. See the exact porosity contents in Table S2 in the Supplementary Materials.

The laser parameter optimization mitigated the porosity, but also revealed the occurrence of cracking in the L-PBF-built H230. At 80 °C, most defects are highly anisotropic and align with the build direction (BD). At 200 °C, these elongated defects were eliminated, and only a few pores are visible in the cross-section. The comparison of the metallography in Figure 3 highlights the efficacy of increasing the pre-heat for mitigating cracking.

To precisely identify the root cause of the cracking problem is difficult and beyond the scope of this study. Clearly, the cracks could originate from various formation mechanisms because the as-built component experiences a complex thermal history in AM which involves multiple remelting and reheating cycles imposed by various scanning patterns. However, the formation of cracks always requires two essential elements—a susceptible microstructure and a mechanical restraint.

In a comprehensive study on the cracking behaviors in the L-PBF-built Haynes 282, Otto et al. [18] found that the $\gamma'$ precipitates preferably formed at the $\gamma/M_{23}C_6$ interfaces. The orientation relationship of the $\gamma/\gamma'/M_{23}C_6$ interfaces is cube-on-cube and fully coherent. They argued that the misfit between the phases imposed residual stress on the coherent boundaries which could in turn assist the formation of micro-cracks. H230 is a solution strengthening superalloy which has low concentrations of the strengthening precipitate-forming elements, e.g., Al and Ti. That said, it is less susceptible to strain-age cracking caused by the $\gamma'$ formation. In fact, Yang et al. [14] discovered that the $\gamma/M_{23}C_6$ interfaces in the L-PBF-built H230 are incoherent, with a mismatch strain of 6%.

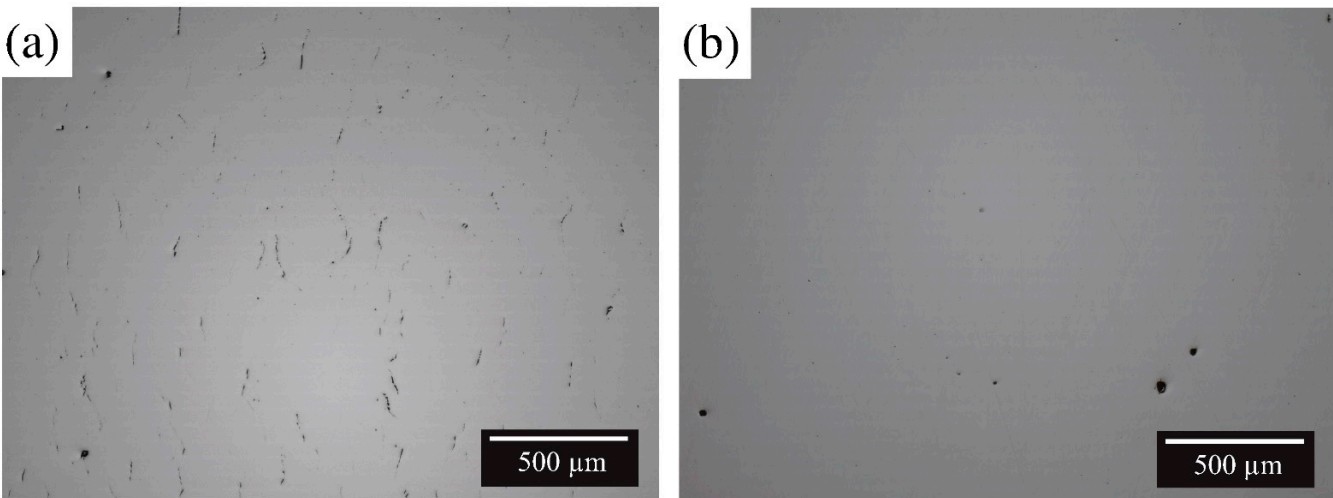

**Figure 3.** The optical micrographs of the polished cross-sections of the L-PBF H230 specimens built using the L2/H2 laser parameter at (**a**) 80 °C and (**b**) 200 °C pre-heat temperatures. The comparison highlights the reduction in cracks in the as-built coupons after increasing the pre-heat temperature. Note that the vertical direction is the build direction.

Additionally, there are two more possible weak links in the microstructure which could be the preferred crack nucleation sites. Partitioning could lead to eutectic formation at the final solidification regions, e.g., the grain boundaries and the centerline of a melt track. Owing to the lower melting point compared to the surrounding microstructures, the eutectics could be selectively remelted and torn during the subsequent laser passes. Secondly, the gaps between the dendritic arms in the mushy zone due to the lack of backfilling could be responsible for the potential solidification cracks.

Interestingly, the recent publications [12,14] on the L-PBF-built H230 did not report any observations of cracks from the metallography. A slight variation of the feedstock composition or the building conditions, e.g., the pre-heat temperature, could have resulted in the different cracking responses. Ernst [31] has reported the cracking problem in a weldability test, which resulted in both liquation and solidification cracks from the H230 weld beads. The study argued that the presence of boron depressed the terminal solidification temperature and increased the wettability of the liquid, which could be detrimental to the cracking problem [32].

The etched microstructure of the as-built H230 in Figure 4a revealed both the cellular/dendritic and the melt pool boundaries. As highlighted, some cracks reside at the centerline of a melt pool which favors crack nucleation at solidification by providing the eutectic composition, concentrating the horizontal tensile stress and limiting liquid backfilling. Other cracks were found to separate the cellular/dendritic domains, as shown in Figure 4b. Some of them have serrated morphology with clearly visible solidification features, e.g., dendrite tips, which could be a sign of solidification cracking. According to the EBSD map in Figure 4c, many cracks are transgranular. On both sides of these cracks at least, the misorientations are relatively small.

The different stress magnitudes in the horizontal ($\sigma_{xx}$) and the build direction ($\sigma_{zz}$) could explain the anisotropy of the cracks. Figure 5a–d shows the maximum stress experienced at each voxel after the last remelt by the subsequent laser passes. At both pre-heat temperatures, more than 90% of the voxels experienced a maximum stress, $\sigma_{xx}$, between 350 MPa and 450 MPa. In contrast, more than 50% of the voxels experienced a maximum stress, $\sigma_{zz}$, below 10 MPa. As shown in Figure 5c,d, only the voxels near the outer surface experienced the higher vertical stresses. That said, the plastic deformation is expected as the maximum stresses are on a par with the yield stress of the material. In addition, for the material away from the outer shell, vertical cracks are more likely to develop. Statistically, even without considering the stress distribution, it is still more probable for cracks to

propagate along the build direction. This is because the epitaxial growth is often dominant in AM solidification. It results in elongated grains, thus introducing anisotropy to the densities of the aforementioned weak links.

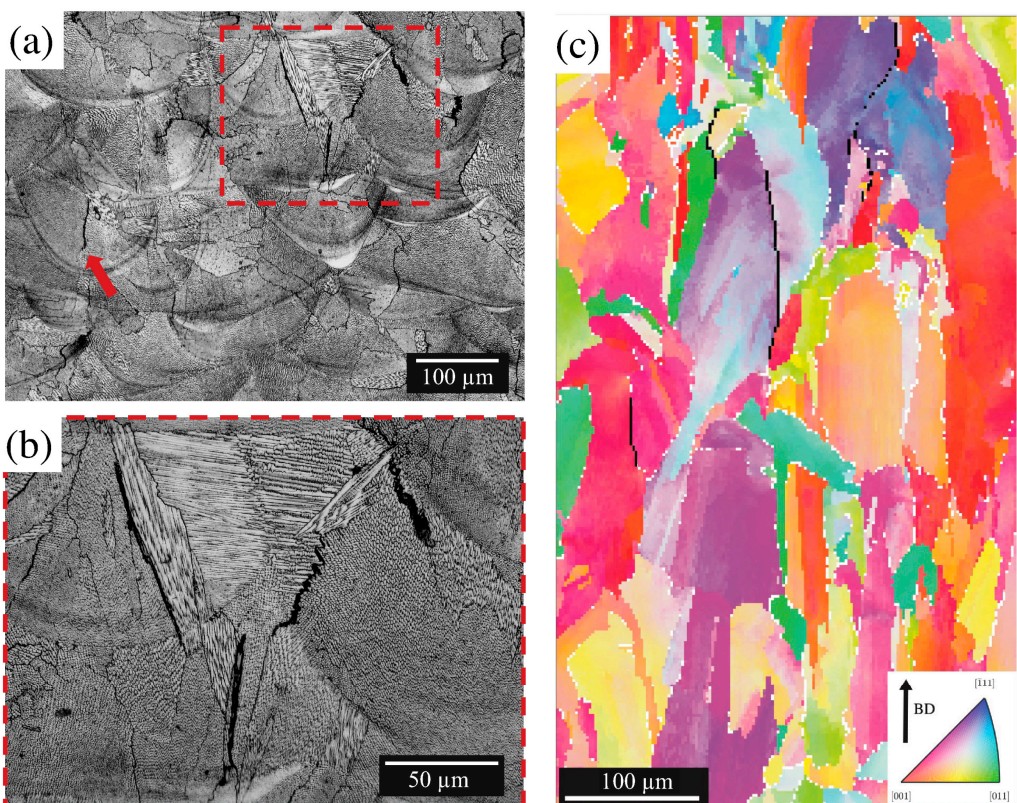

**Figure 4.** An optical micrograph in (**a**) shows the etched microstructure of the as-built H230 printed at 80 °C pre-heat temperature, and (**b**) a magnified view of the marked area in (**a**) highlights the cracks at the boundaries of cellular domains. Note that the red arrow in (**a**) indicates a crack at the centerline of a melt pool. An EBSD map in (**c**) reveals some transgranular cracks. The EBSD map was captured at an accelerating voltage of 30 kV and a step size of 1 μm.

As shown in Figure 5e, the simulations suggest that the influences of increasing the pre-heat temperature on the stress magnitudes and distributions are insignificant. One speculation is that the difference between the two pre-heat temperatures was too small to impact the global cooling rates and thermal gradients. Both 80 °C and 200 °C are below the temperature limit at which stress relief may occur given the short duration of the printing process. Figure 5f shows the reductions of the maximum stresses at each voxel after increasing the pre-heat from 80 °C to 200 °C. The symmetrical distributions confirmed that the effect is minor in terms of eliminating stress hot spots despite the slightly positive skewness of the distributions. This observation obviously contradicts the effective porosity reduction observed in the experiments.

Finally, the solidification features found on the cross-sections and the minor stress reduction due to the higher pre-heat could not prove when exactly the cracks formed and propagated with respect to the cooling sequence. A mixture of mechanisms could contribute to the crack formation at the same time. However, this evidence did suggest that the solidification and liquation cracking is more susceptible given no $\gamma'$ precipitation in the H230. It also emphasized the importance of the weak links at liquidus, of which the severity could be changed significantly by the different pre-heats. Future works, including analyzing the boron contents in the feedstock, could be helpful in further understanding the solidification behaviors.

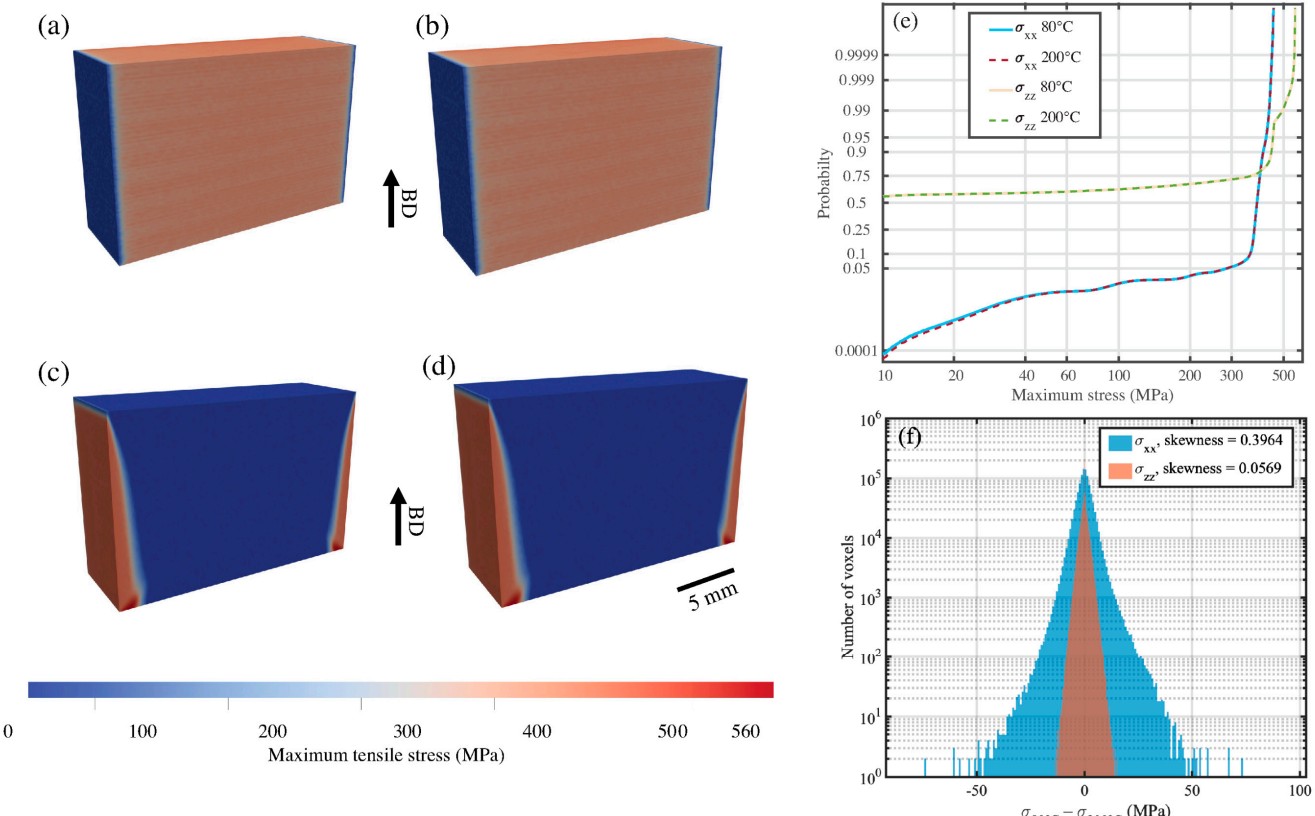

**Figure 5.** The finite element simulated stress spatial distributions show the maximum stresses ($\sigma_{xx}$, $\sigma_{zz}$) experienced by each voxel at (**a**,**c**) 80 °C and (**b**,**d**) 200 °C pre-heat temperatures, as well as the corresponding probability distributions in (**e**). The histograms in (**f**) summarize the stress difference ($\sigma_{80°C} - \sigma_{200°C}$) at each voxel.

### 3.2. Heat-Treated H230

The H230 specimens, including the mechanical coupons, were annealed at 1200 °C for 2 h, followed by water quench. Figure 6a shows the columnar grains along the build direction in the specimen prior to the heat treatment. The <100> pole figure in Figure 6b suggests that the as-built specimen has a rotated cube texture, which is frequently reported in L-PBF-built superalloys [33,34]. The heat treatment caused insignificant recrystallization. As shown in Figure 6c,d, the grain morphology and the microstructural texture were preserved. In fact, the texture strength increased after the heat treatment. Location bias due to the limited characterization areas could have contributed to this increment; meanwhile, the heat treatment stress relieved the specimen. The orientation gradient within the same grain could be reduced, and this in turn increased the texture strength.

In Figure 6e, the channeling contrast reveals the grain structures in the BSE image, but the carbides at the cellular boundaries can only be visible at higher magnification, as shown in Figure 6f. These fine carbides are <100 nm in size. The heat treatment brought out the grain boundary carbides which were not present in the as-built condition. As shown in Figure 6g, the grain boundary carbides (~2 μm) are significantly coarser than the cellular boundary carbides (~200 nm) even though they were coarsened by the heat treatment. The EDS spectrum (Figure S2) shows that the grain boundary carbides are enriched in W. In a comparable study, Yang et al. [14] found that an annealing treatment at a lower temperature (980 °C for 1 h, followed by furnace cool) could transform the Cr-enriched $M_{23}C_6$ carbide to the W- and Mo-enriched $M_6C$ carbide in the L-PBF-built H230. No cracks were found at the interfaces between the γ matrix and the newly precipitated carbides. This implies that ductility-dip cracking and strain-age cracking are irrelevant, at least for the cooling rate

during the water quench, which is several orders of magnitude slower compared to the cooling rate during solidification.

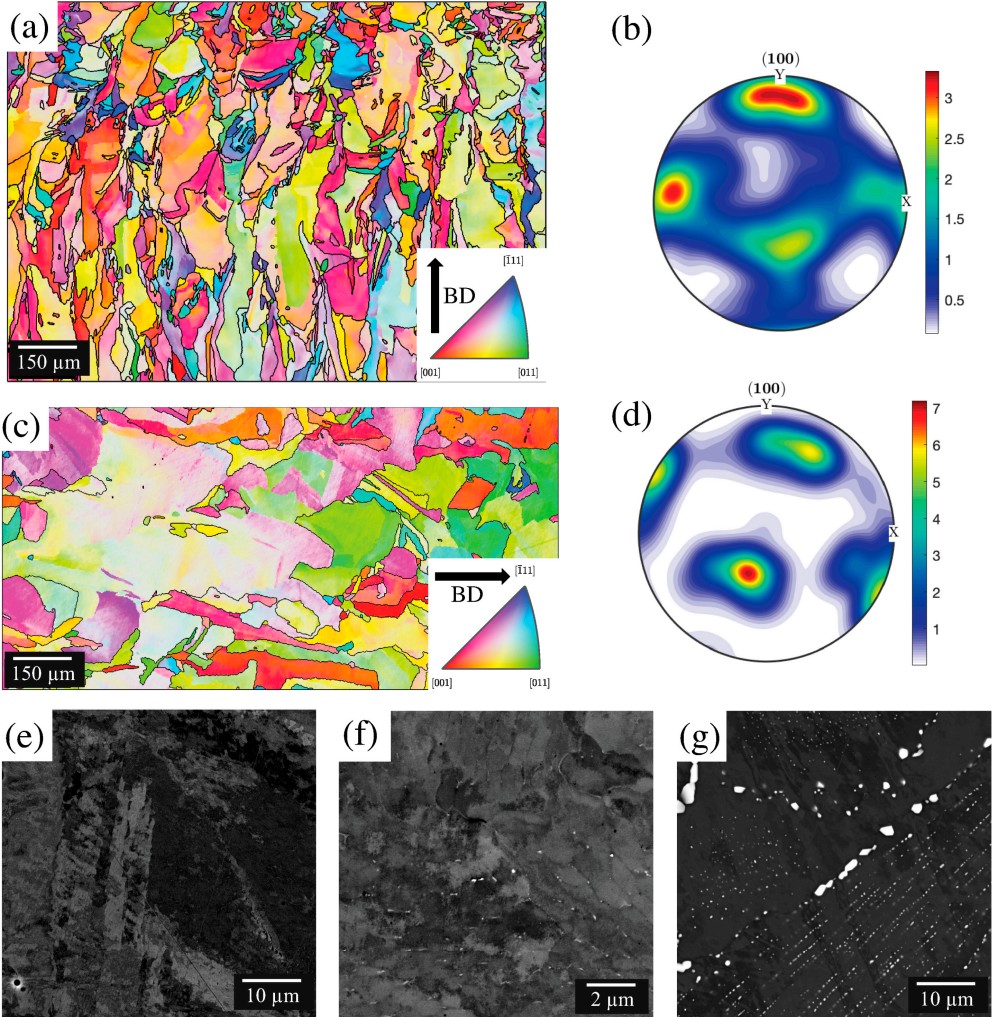

**Figure 6.** The crystal orientation maps and the pole figures show the microstructural textures of the L-PBF-built H230 at (**a**,**b**), the as-built, and (**c**,**d**) the annealing conditions. The EBSD data were captured using an accelerating voltage of 30 kV, and a step size of 1 μm. The BSE images highlight the carbide coarsening process from (**e**–**g**) introduced by the annealing treatment.

### 3.3. Room Temperature Tensile Properties

Four H230 specimens were tested at room temperature with the tensile direction parallel to the build direction. Table S3 summarizes the quantitative results of each specimen. As shown in Figure 7, the stress-strain responses of the AM H230 are consistent. Note that the strain was only recorded until 0.015 to protect the extensometer. The average modulus, the average 0.2% yield stress, the average ultimate tensile stress (UTS), and the elongation are 181.7 GPa, 438.7 ± 11.4 MPa, 858.4 ± 16.6 MPa, and 43.31 ± 3.1%. For the wrought H230, as shown in Table S3, these values are 353 MPa, 852 MPa, and 46% [2]. Both the UTS and the elongation are within a single standard deviation, while the yield stress of the AM H230 is higher than its wrought counterpart. That said, the performance, i.e., the ductility and strength, of the AM H230 matches that of the wrought material in high strain rate room temperature testing.

The high ductility suggests no excessive carbide coarsening during the heat treatment. This is also evidenced by Figure 6g. In addition to the water quenching method, four specimens were annealed at 1165 °C for 3 h, followed by furnace cool. Table S4 shows the tensile results of the slowly cooled H230. Furnace cooling resulted in carbide coarsening.

As a result, the specimens are much stronger, i.e., the yield stress is 480.6 ± 2.6 MPa and the UTS is 904.1 ± 4.6 MPa, but they are more brittle, i.e., the elongation is 33 ± 1.4%.

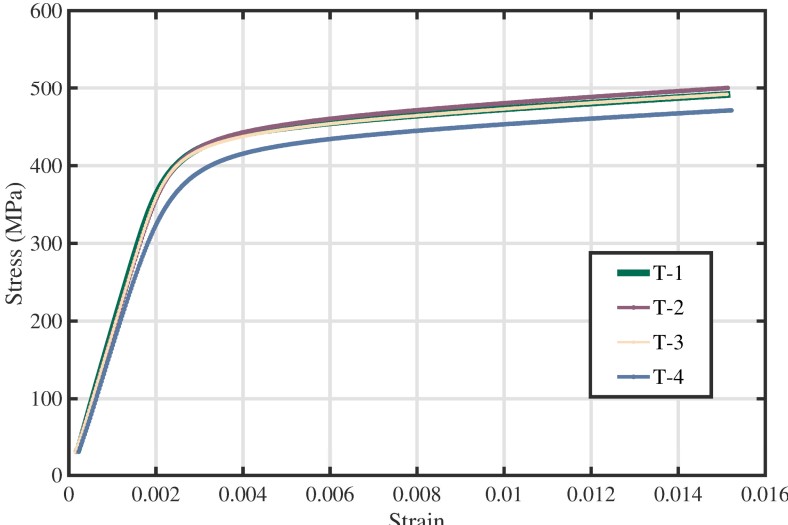

**Figure 7.** Room temperature stress-strain curves of the L-PBF H230 in the heat-treated condition. Note that the extensometer was removed at a strain of 0.015 to prevent damage.

Figure 8a also suggests that ductile fracture is the primary fracture mode, based on the dimple pattern observed on the fracture surface. Printing defects were occasionally found on the fracture surfaces, but no evidence suggests that these defects caused crack initiation. Figure 8b shows an example of a spherical pore, which is speculated to be a keyhole pore due to its morphology. The smooth side wall decorated with carbide precipitates suggests that the defect formed at solidification.

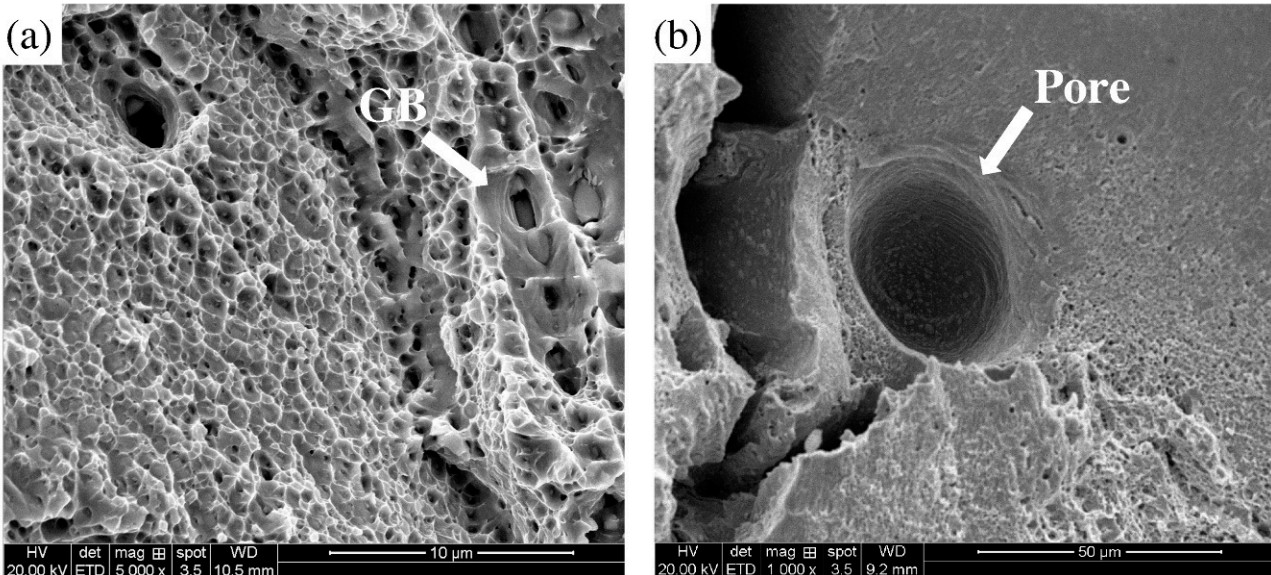

**Figure 8.** The SEM images show (**a**) a grain boundary and the dimple pattern and (**b**) a keyhole pore on the fracture surfaces of the L-PBF H230 tensile specimens.

### 3.4. Creep Properties at 760 °C and 816 °C

The primary goal of the creep test was to investigate whether the creep properties of the L-PBF H230 match its wrought counterpart. This evaluation is an important step in demonstrating the feasibility of AM as an alternative fabrication route for critical components for high-temperature applications. Four specimens were tested in each of the two

test conditions—(a) 760 °C and 100 MPa and (b) 800 °C and 121 MPa. The detailed creep test results are summarized in Tables S5 and S6.

As shown in Figure 9a, no rupture occurred after testing at 100 MPa and 760 °C for 500 h. The total creep strains vary from 0.72% to 3.5%. C-L1 is the outliner, in which the total creep strain substantially deviated from the other three samples. Unlike C-L2–C-L4, which maintained stable creep rates, the creep of C-L1 accelerated after 250 h of testing. Only C-L1 deflected from the minimum creep rate, while the others were still approaching it, as shown in Figure 9b. One speculation about this deviation is that the creep testing is sensitive to the local density of the grain boundaries in the printed materials. At 300 h, the creep damage at the highly concentrated grain boundaries exceeded the critical threshold and resulted in the tertiary creep behavior. The creep strains of C-L2–C-L4 are very similar. C-L3 showed the highest creep strain of 0.727% among the three specimens after 300 h of testing. These three sets of data are comparable to the creep properties of the wrought H230 tested by Boehlert et al. [6] with the same conditions. The only difference was that the specimen in Boehlert's study was solution annealed at 1149 °C, followed by water quench.

At 121 MPa and 816 °C, all four specimens ruptured within 41 h. These numbers fall between the rupture life reported by Haynes International [2], as shown in Table S6 for the wrought H230 which is >100 h, and the 18 h rupture life of the wrought H230 tested at a lower temperature (800 °C) and a lower stress (100 MPa), as reported by Pataky et al. [7]. The different rupture lives reported in the literature suggest that significant variations exist in the wrought materials. Particularly when the testing temperature is high, the creep life can be more sensitive to the microstructure and the defect size. This is because the secondary creep phase would dominate the creep process in terms of duration at lower testing temperatures, introducing a critical level of creep damage and causing creep failure, while at higher temperatures its importance is reduced as the tertiary creep is easily activated at smaller strains. As a result, larger variations in the creep properties obtained at higher temperatures were anticipated.

C-H1–C-H4 ruptured at total strains between 30% and 40%, and the total strains were not rupture life-dependent. The creep entered the tertiary phase after 20 h of testing, as shown in Figure 9d. Compared to the results at 760 °C, no obvious creep rate reduction occurred before reaching the minimum. This suggested that the rate-limiting creep mechanisms are different at the two temperatures. C-H4 is the outlier among the four samples. It showed higher creep rates from the initial stage of the test. However, the creep results reported by Pataky et al. [7] overlap with the curve of C-H4, as shown in Figure 9c. That said, C-H1–C-H3 consistently outperformed its wrought counterparts.

At 760 °C, the L-PBF H230 has low minimum creep rates between $10^{-2}$%/h and $10^{-3}$%/h, which means that the printed parts should change dimensions rather slowly once the initial transient has exhausted the hardening. At 816 °C, the minimum creep rates are between 1%/h and $10^{-1}$%/h. This implies that H230 is not suitable for this combination of temperature and stress. In terms of minimum creep rates, Figure 9e shows that the L-PBF and the wrought H230 are comparable. The minimum creep rates at 760 °C and 816 °C are all below the fitting curves for the wrought materials tested at similar conditions. Pataky et al. [7] suggested that the rate limiting mechanism at 800 °C is dislocation climb as the exponent n is around 5. The formation of carbides could pin the grain boundary and develop a stress field to prevent grain boundary sliding; thus, the creep rate is reduced. As shown in Figure 6g, the heat treatment encouraged the formation of grain boundary carbides which could have benefited the creep performance of the AM components.

As in the tensile specimens, a dimple pattern dominates the fracture surfaces of the creep specimens which suggests that ductile fracture was the primary failure mode. Creep damage mostly appears in the form of grain boundary separation. An example highlighted in Figure 10a shows a smooth interface decorated with grain boundary carbides. However, no evidence was found to show that any printing defects directly contributed to the crack nucleation.

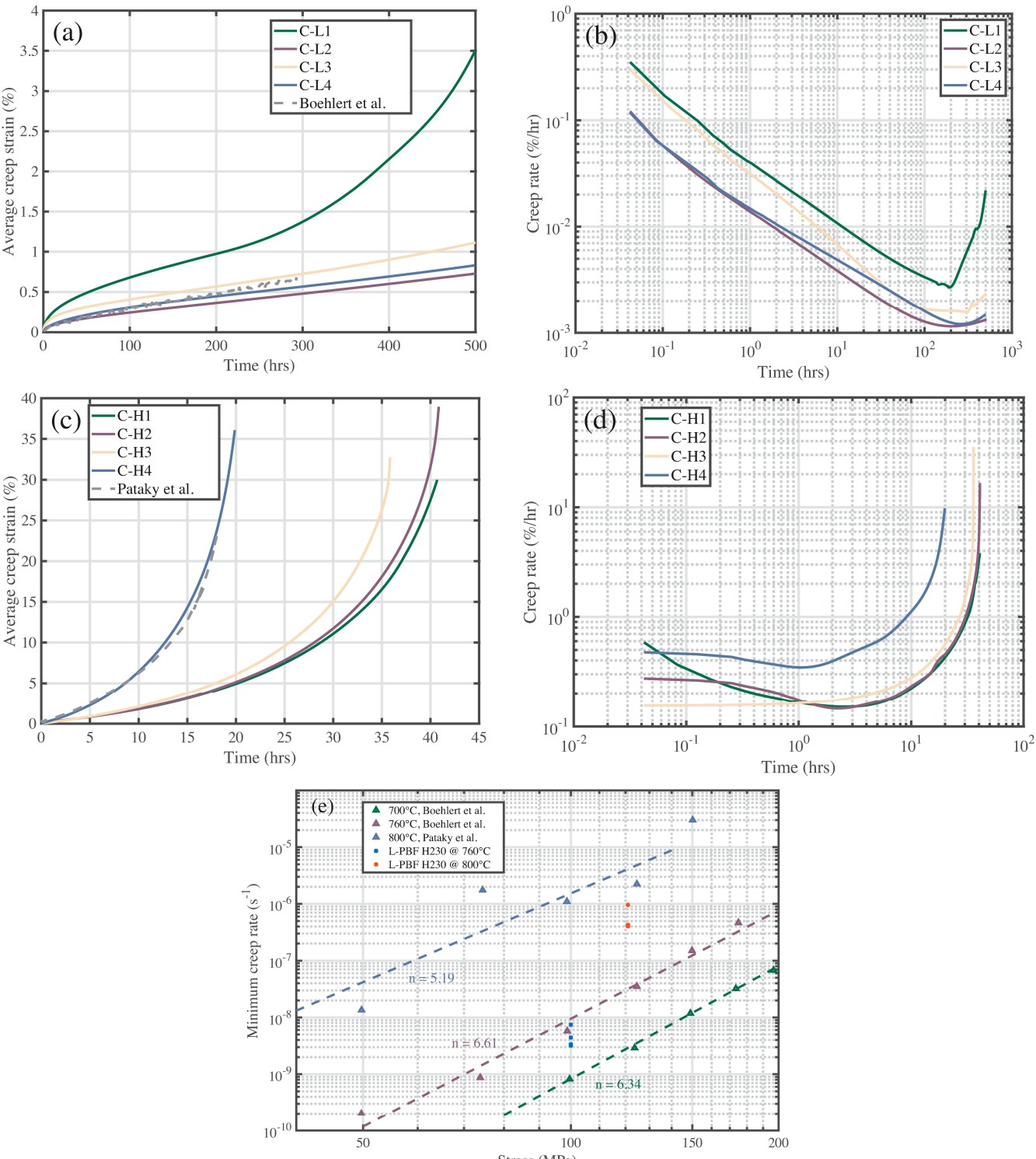

**Figure 9.** Creep strains and the creep rates as a function of time measured at the test conditions of (**a**,**b**) 760 °C and 100 MPa and (**c**,**d**) 816 °C and 121 MPa. The reference data include the wrought H230 tested by Boehlert et al. [6] at 760 °C and 100 MPa and Pataky et al. [7] at 800 °C and 100 MPa. The minimum creep rates of the L-PBF Haynes 230 are compared with its wrought counterparts, as reported by the literature in (**e**).

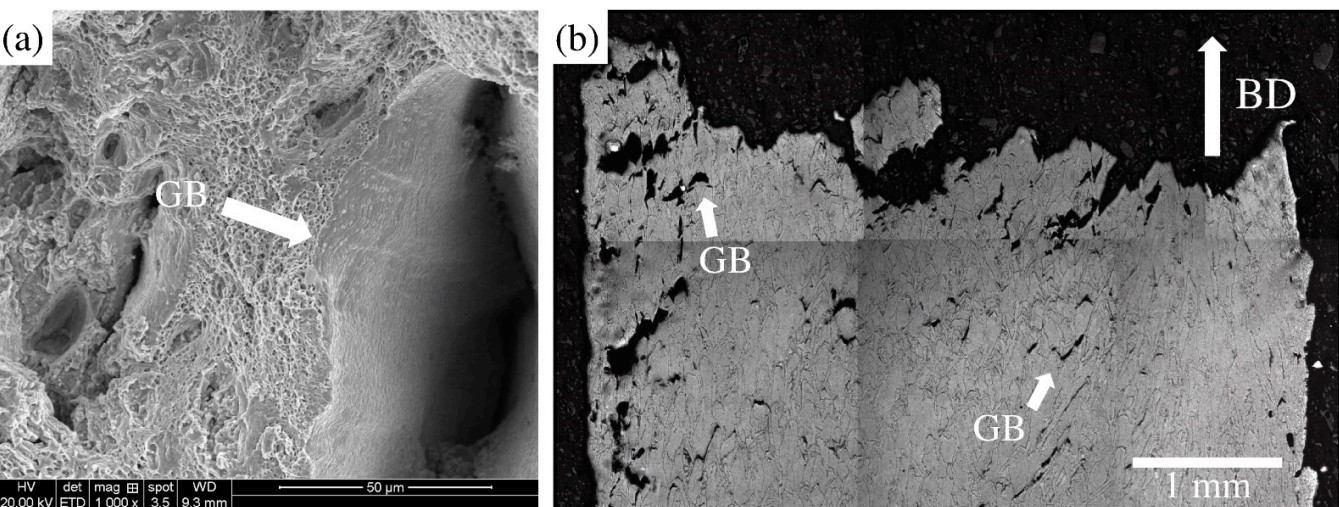

**Figure 10.** A SEM image and an optical image show (**a**) the fracture surface and (**b**) the vertical cross-section of a ruptured creep specimen tested at 816 °C and 121 MPa. Some grain boundary (GB) separations are highlighted in the figure.

## 4. Conclusions

This study revealed that the optimal process space for the L-PBF H230 had energy densities between 67.5 J/mm$^3$ and 120.1 J/mm$^3$. Nearly fully dense coupons were successfully fabricated in an EOS M290 machine after optimizing the process parameters.

In addition to porosity, cracks were identified as another barrier for producing structurally sound H230 via AM. An increased pre-heat temperature from 80 °C to 200 °C sufficed to eliminate cracking under the conditions used. The cracks were highly oriented and parallel to the build direction because of the high horizontal principal stresses. In addition, they were often transgranular and propagated along the centerline of a melt track. A finite element stress simulation suggested that the pre-heat temperature had limited impacts on the maximum stress experienced by each location. At both pre-heat temperatures, the maximum horizontal stresses were between 350 MPa and 450 MPa. That said, the cracks were more likely to nucleate near the liquidus, i.e., solidification cracking and liquation cracking, as some solidification features were also observed, such as dendrite tips. The higher pre-heat temperature could have facilitated better backfilling to prevent crack nucleation. The outcome in this study suggests that changing the pre-heat temperature could be a viable optimization strategy for alloys prone to cracking and should attract more attention in the effort to improve the printability of alloys.

This study also performed room temperature tensile tests and creep tests at two different temperatures and stresses on the annealed L-PBF H230 coupons. The annealing treatment followed by the water quench limited the undesired carbide coarsening and the alteration of the microstructural texture. As a result, the L-PBF H230 performed comparably to the wrought counterparts and displayed a balance between ductility and strength in the mechanical testing. The sizes/concentrations of the printing defects were successfully controlled at a low level, which was believed to have limited the influences on the mechanical properties. That said, the existing data of the wrought H230 high-temperature properties should be directly applicable when evaluating the high-temperature performance of the AM-built H230 components.

**Supplementary Materials:** The following supporting information can be downloaded at: https://www.mdpi.com/article/10.3390/met12081380/s1, Figure S1: A SEM image and the corresponding size distribution of the H230 powder feedstock; Figure S2: A SEM image and the corresponding EDS spectrum of a grain boundary carbide in a L-PBF-built H230 specimen annealed at 1200 °C for 2 h followed by water quench. The red dot highlights the location where the EDS spectrum was acquired; Table S1: H230 material properties [2] used in the finite element stress simulations;

Table S2: The raster process parameters of the testing coupons and the corresponding porosity contents. Note that all specimens share the same contour parameters. The marker 'x' indicates a failed part due to excessive energy input and extreme scanning velocity. The parameters for the mechanical coupons are highlighted in **bold**; Table S3: Summary of the tensile properties of the L-PBF H230 at the annealing condition (1200 °C for 2 h followed by water quench) and the wrought H230 reported Haynes International, Inc. [2]; Table S4: Summary of the tensile properties of the L-PBF H230 at the stress relieved condition (1165 °C for 3 h followed by furnace cool) and the wrought H230 reported Haynes International, Inc. [2]; Table S5: Summary of the creep properties of the L-PBF H230 at the annealing condition (1200 °C for 2 h followed by water quench) and the wrought H230 reported Haynes International, Inc. [2] tested at 760 °C and 100 MPa; Table S6: Summary of the creep properties of the L-PBF H230 at the annealing condition (1200 °C for 2 h followed by water quench) and the wrought H230 reported Haynes International, Inc. [2] tested at 816 °C and 121 MPa.

**Author Contributions:** Z.W.: methodology, validation, formal analysis, investigation, writing—original draft, data curation, visualization; S.R.Y.: methodology, validation, formal analysis, investigation, writing—review and editing, data curation; J.S.: writing—review and editing, formal analysis, data curation. N.L.: writing—review and editing; A.D.R.: conceptualization, writing—review and editing, project administration, supervision, funding acquisition. All authors have read and agreed to the published version of the manuscript.

**Funding:** This material is based upon work supported by the U.S. Department of Energy's Office of Energy Efficiency and Renewable Energy (EERE) under the Solar Energy Technologies Office Award Number DE-EE0008536.

**Institutional Review Board Statement:** Not applicable.

**Informed Consent Statement:** Not applicable.

**Data Availability Statement:** The authors confirm that the data supporting the findings of this study are available within the article and its Supplementary Materials.

**Acknowledgments:** Carpenter Technologies is thanked for providing the Haynes 230 powder used in this work. The authors acknowledge the support of the NextManufacturing Center and the use of the Materials Characterization Facility, supported by grant MCF-677785, at Carnegie Mellon University. Specifically, Todd Baer is acknowledged for sample fabrication and preparation. Z.W. was supported under the auspices of the U.S. Department of Energy by Lawrence Livermore National Laboratory under contract DE-AC52-07NA27344. Release number #LLNL-JRNL-836489.

**Conflicts of Interest:** The authors declare no conflict of interest.

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
