# Peer review of "Study of the Printability, Microstructures, and Mechanical Performances of Laser Powder Bed Fusion Built Haynes 230"

_metals, doi:10.3390/met12081380_

Round 1

Reviewer 1 Report

This work investigated the printability, microstructures, and mechanical performances of laser powder bed fusion-built Haynes 230. It is an important characterization of the widely used Haynes 230 superalloy at high temperatures and not yet vastly adopted as 3D-printed material compared to its wrought counterpart. In particular, the work highlights the effect of pre-heat temperature on reducing porosity and crack formation, which might be of great interest for the use of the alloy in aerospace applications. 

The manuscript could be accepted for publication in metals after considering the following points:

(1) Page 8, lines 286-288: The results of finite temperature simulations ''suggest that the influences of increasing the pre-heat temperature on the stress magnitudes and distributions are insignificant''. The authors are invited to elaborate more on the reasons behind the identical stress curves at 80°C and 200°C shown in Fig. 5(e).

(2) The scales of optical micrographs and BSE images should be carefully rechecked. In particular, those of Fig. 4(c) and Fig. 6(g).

Minor comments:

(1) " Error! Reference source not found" message has occurred many times after building the PDF file, which may be related to the removal of caption or content that is " referenced" somewhere else in the file. 

(2) Page 2, lines 56-57: " they found that H230 maintained its the tensile properties and the microstructure after a long exposure at 850°C". The unnecessary "the", repeated twice, should be omitted.

Reviewer 2 Report

The paper entitled "Study of The Printability, Microstructures, and Mechanical Performances of Laser Powder Bed Fusion Built Haynes 230" has been presented for review. 

1. you have issues with the references in the text so the correctness of the references could not be verified. (For example lines 133-134; 139-140, etc.);

2. Section 2 and supplementary material. From my point of view, Fig. S1 has a too low magnification to observe the morphology of the powder. Moreover, the PSD data has not been presented. 

3. Lines 146-150 - why did you decide to remove the coupons from the substrate before post-heat treatment? Higher residual stress could be expected in such a case. 

4. Fig. 1b - please specify according to which standard the tensile/creep coupons were prepared.

5. Lines 159-160 - "The parameters for the mechanical coupons are highlighted in bold." - that means that only sample #H2 was used for mechanical tests, wasn't it?

6. For Introduction I recommend to view the following additional papers:

https://doi.org/10.1016/j.msea.2012.09.017 

https://doi.org/10.3390/ma14040909

https://doi.org/10.3390/ma14174792

7. Figures' numbering is not correct, please fix it. The same to tables. There are no references to tables in the text. 

8. I did not find a comparison of tensile results with the data of wrought alloy.

9. In conclusion please, add a more clear statement on how promising were results for additively manufactured coupons were in comparison with the wrought alloy data. 

Reviewer 3 Report

The main goal of the submitted paper refers to studies concerned with printability, microstructure analysis and evaluation of mechanical properties of Haynes 230 superalloy manufactured with the use of the LPBF system. The paper is well written. Adopted by Authors structure of the manuscript is following commonly used scientific paper standards.  Nevertheless, as a reviewer I have the following remarks:
1)    Missing references in the text of the manuscript. Dear Authors, please read the paper again twice before submission to avoid these types of errors.
2)    Remark to section Tensile and creep tests
-    You wrote that tensile tests were carried out with the as-built surface finish. Furthermore, you wrote that you have used an extensometer to identify the value of deformation. I’m wondering if it is a proper way because of the surface roughness. I recommend adding a comment on why you didn’t use an additional machining process to reduce the surface roughness before tests.
-    I recommend changing the unit of strain rate. Please change this value from 0.005 in./in./min to -  [1/s-1].
3)    Remark to section 3.1 Printability of H230
-    In my opinion, Fig.2 will be more readable if you will add another chart e.g. Fig.2B, which will illustrate the value of porosity as a function of energy density (identified for a particular set of parameters). This chart can be useful to define the operating windows. Of course, treat my comment as a suggestion.
4)    Remarks related to computer simulations carried out in the Ansys Additive module.
-    There is a lack of adopted material properties in numerical simulations. Please add this information in this section. Furthermore, there is a lack of any information related to adopted initial boundary conditions (e.g parameters applied to define the heat distribution between part and powder bed).
5)    Poor quality of Fig. 6a and Fig.6b. The scale on both figures is almost invisible. Please use the higher resolution of images.
6)    Remarks on section “Room temperature tensile properties”
-    If the Authors used the extensometer why they didn’t identify the young modulus for studying H230 superalloy? This parameter will be useful for other researchers.
1)    Remarks on section Conclusions:
-    Please add some information related to the potential usage of your results for other researchers. Please add some comments referring to AM process. What other researchers should do to eliminate or minimize the effects of microstructural material imperfections if they will use a similar type of superalloy?

Round 2

Reviewer 1 Report

The authors have answered all my comments. The manuscript could be accepted in the present form.

Reviewer 2 Report

All my comments were answered